# Gear Shifting in Biological Energy Transduction

**DOI:** 10.3390/e25070993

**Published:** 2023-06-28

**Authors:** Yanfei Zhang, Hans V. Westerhoff

**Affiliations:** 1Synthetic Systems Biology and Nuclear Organization, Swammerdam Institute for Life Sciences, University of Amsterdam, 1098 XH Amsterdam, The Netherlands; y.zhang2@uva.nl; 2Department of Molecular Cell Biology, Faculty of Science, Vrije Universiteit Amsterdam, De Boelelaan 1085, 1081 HV Amsterdam, The Netherlands; 3School of Biological Sciences, Faculty of Biology, Medicine and Health, University of Manchester, Oxford Road, Manchester M13 9PL, UK; 4Stellenbosch Institute for Advanced Study (STIAS), Wallenberg Research Centre at Stellenbosch University, Stellenbosch 7600, South Africa

**Keywords:** gear shifting, thermodynamics, non-equilibrium thermodynamics, phenomenological stoichiometry, cell growth, ATP synthesis

## Abstract

Confronted with thermodynamically adverse output processes, free-energy transducers may shift to lower gears, thereby reducing output per unit input. This option is well known for inanimate machines such as automobiles, but unappreciated in biology. The present study extends existing non-equilibrium thermodynamic principles to underpin biological gear shifting and identify possible mechanisms. It shows that gear shifting differs from altering the degree of coupling and that living systems may use it to optimize their performance: microbial growth is ultimately powered by the Gibbs energy of catabolism, which is partially transformed into Gibbs energy (‘output force’) in the ATP that is produced. If this output force is high, the cell may turn to a catabolic pathway with a lower ATP stoichiometry. Notwithstanding the reduced stoichiometry, the ATP synthesis flux may then actually increase as compared to that in a system without gear shift, in which growth might come to a halt. A ‘variomatic’ gear switching strategy should be optimal, explaining why organisms avail themselves of multiple catabolic pathways, as these enable them to shift gears when the growing gets tough.

## 1. Introduction

Thermodynamics is a branch of chemical physics that studies the relationship between heat, energy, and work and how these quantities affect the behavior of matter [1]. Thermodynamics encompasses two branches: equilibrium thermodynamics and non-equilibrium thermodynamics. Equilibrium thermodynamics deals with the macroscopic behavior of systems that are at equilibrium. It mainly investigates how variables such as entropy, volume, and ligand binding change with respect to the changes in the equilibrium system’s surrounding temperature, pressure and ligand concentrations through reversible energy, heat, and material transfer [2]. In contrast, non-equilibrium thermodynamics pertains to systems that are not in a state of thermodynamic equilibrium and provides a framework for understanding and predicting the behavior of such systems [3,4,5,6]. In equilibrium systems, net process rates are zero. For their persistence, living systems need to engage in repair and maintenance processes with much additional benefit from their ability to auto-replicate. Accordingly, living systems require non-equilibrium processes and consequently, much of their analysis requires non-equilibrium thermodynamics. The original non-equilibrium thermodynamics [7,8,9] were phenomenological, treating systems as black boxes connecting output with input processes. Therefore, its conclusions were independent of internal structural characteristics and mechanisms. Mosaic non-equilibrium thermodynamics adds mechanism to the thermodynamics and has been applied to ion transport and biological energy-transducing systems [10].

Non-equilibrium thermodynamics has many applications in biology, including in cellular metabolism [10,11,12], cell signaling [13,14], protein folding [15,16], neural activity [17,18], and self-organization and molecular evolution [7,19,20]. In the present paper, we focus on a phenomenon that is emerging in biology, that may be equivalent to ‘gear shifting’ [21], but is rarely recognized as such. Gear shifting is a common concept in driving, where the driver (or the car itself) shifts gear depending on the road conditions. When driving on a flat highway, the highest gear will give the highest speed, while when driving uphill, shifting to a lower gear may increase the speed, as it is necessary to provide enough force to the wheels: the optimal gear is not always the highest. Living organisms may exhibit similar gear shifting. *Clostridium ljungdahlii*, for instance, contains alternatives in its genome of redox reactions that produce different ATP/acetate ratios and have been proposed to mediate gear shifting [22]. *Paracoccus denitrificans* and *Escherichia coli* avail themselves of various terminal oxidases enabling gear shifting in their proton pumping activity [23,24]. *Saccharolobus solfataricus* contains the non-phosphorylating glyceraldehyde-3-phosphate dehydrogenase (GAPN) [25,26], which catalyzes the direct oxidation of glyceraldehyde-3-phosphate to 3-phosphoglycerate by NADP. The enzyme bypasses adenosine 5-triphosphate (ATP) formation by substrate-level phosphorylation of ADP via phosphoglycerate kinase (PGK), and the choice between the two routes (GPAN or GPADH-PGK) gives the organism the ability to switch gears.

In this paper, we develop the theory behind gear shifting in cell growth. We employ both mosaic and phenomenological non-equilibrium thermodynamics. Specifically, we examine how gear shifting may benefit an organism when growth becomes hard thermodynamically.

## 2. Materials and Methods

This section describes the computations leading to the figures in this paper in a technical manner. The underlying theory is given in the Results section.

### 2.1. Variation of the Force Ratio May Induce Catabolic Gear Shifting

A system consisting of a pathway 1 and a pathway 2 with different ATP stoichiometries ($n_{1}$ = 1, $n_{2}$ = 2) was considered here. The thermodynamic driving force for the reaction is the Gibbs energy drop across the catabolic reaction, denoted by $\Delta G_{c}$. The ATP synthesis flux is written as $-J_{p}$ and the total (see below) catalytic capacity of the catabolic ATP synthesis reactions as L. φ represents the fraction of that catalytic capacity that resides in pathway 2, and $n_{1}$ and $n_{2}$ represent the ATP synthesis stoichiometries of the two pathways. X>0 is the counteracting thermodynamic force ratio, i.e., the ratio of the Gibbs energy of ATP synthesis to the Gibbs energy of catabolism. We simulated the dependencies of the normalized ATP synthesis flux $-J_{p}/\left(L\cdot \Delta G_{c}\right)$ (calculated using equation $-J_{p}/\left(L\cdot \Delta G_{c}\right)=\left(1-\varphi \right)\cdot n_{1}\cdot \left(1-n_{1}\cdot X\right)+\varphi \cdot n_{2}-\varphi \cdot {\left(n_{2}\right)}^{2}\cdot X$), the ratio of the fluxes through pathway 1 ($J_{p1}/J_{p})$ and pathway 2 ($J_{p2}/J_{p})$, the variable flux ratio stoichiometry n ($-J_{p}/J_{c}$), and the thermodynamic efficiency η (calculated using equation $\eta =-J_{p}/J_{c}\cdot X$), all on force ratio X. In these simulations, both the catalytic capacity of catabolism L, and the Gibbs energy of catabolism $\Delta G_{c}$ were set to 1, the catalytic activity of pathway 2 as compared to the total activity *L* (φ) was set to 0.2. The simulation was for the absence of the specific uncoupling mechanism $\left(L_{p}^{\ell }=0\right).$ $-J_{p1}$ was calculated using equation $-J_{p1}=\left(1-\varphi \right)\cdot n_{1}\cdot L\cdot \Delta G_{c}\cdot \left(1-n_{1}\cdot X\right).$ The total ATP synthesis flux was calculated using equation $-J_{p}$ = $\left(\left(1-\varphi \right)\cdot n_{1}\cdot \left(1-n_{1}\cdot X\right)+\varphi \cdot n_{2}-\varphi \cdot {\left(n_{2}\right)}^{2}\cdot X\right)\cdot \;L\cdot \Delta G_{c}$, while $-J_{p2}$ = $-J_{p}-\left(-J_{p1}\right)$. $J_{c}$ was calculated using equation $J_{c}=\left(\left(1-\varphi \right)\cdot \left(1-n_{1}\cdot X\right)+\varphi \cdot \left(1-n_{2}\cdot X\right)\right)\cdot L\cdot \Delta G_{c}$.

### 2.2. Gear Shifting Simulations

In this section, we first simulated the production of the normalized ATP synthesis flux and the flux ratio stoichiometry n at various force ratios (X) at three values for the phenomenological stoichiometry Z (0.5, 1, 2). In these simulations, the degree of coupling q was set to 0.9. The normalized total ATP synthesis flux $-J_{p}/\left(L\cdot \Delta G_{c}\right)$ was calculated as the sum of three. The relation between flux ratio stoichiometry and normalized ATP synthesis flux for each of the three different phenomenological stoichiometries Z was studied by plotting the flux ratio stoichiometry *n* verse normalized ATP synthesis flux $-J_{p}/\left(L\cdot \Delta G_{c}\right);$ at each value for the phenomenological stoichiometry Z was calculated by the equation $-J_{p}/\left(L\cdot \Delta G_{c}\right)\;$= $q\cdot Z\cdot \left(1-\frac{Z}{q}\cdot X\right)$. 

### 2.3. Simulation of ATP Synthesis Flux through a Dual Pathway

We simulated ATP synthesis flux $-J_{p}/\left(L\cdot \Delta G_{c}\right)\;$($n_{1}\cdot \left(1-\varphi \right)\cdot n_{1}\cdot \left(1-n_{1}\cdot X\right)+\varphi \cdot n_{2}-\varphi \cdot {\left(n_{2}\right)}^{2}\cdot X$) through pathway 1 and pathway 2 with two different ATP stoichiometries when varying the force ratio X. In these simulations, the ATP stoichiometries of pathway 1 and 2 were taken to equal 1 and 3, respectively. Both the catalytic capacity of catabolism L and the Gibbs energy of catabolism $\Delta G_{c}$ were 1, and there was no specific uncoupling mechanism $\left(L_{p}^{\ell }=0\right).$ φ is the catalytic activity of pathway 1 relative to the total catalytic activities of pathways 1 and 2. The simulations were done for four values of φ (1, 0.375, 0.25, 0). In these simulations, both the phenomenological stoichiometry Z and the degree of coupling q varied as functions of φ.

### 2.4. Discontinuous Optimal Gear Shifting

We simulated ATP synthesis $-J_{p}\left(=\left(1-\varphi \right)\cdot n\cdot L\cdot \Delta G_{c}\cdot \left(1-n\cdot X\right)\right)$ as function of counteracting force ratio at four different stoichiometries *n*, i.e., 1, 2, 3, and 4. In these simulations  φ, the catalytic capacity of catabolism *L* and the Gibbs energy of catabolism $\Delta G_{c}$ were set to 0, 1, and 1, respectively. The total ATP synthesis flux $J_{p\;total}$ is the sum of the four ATP synthesis $J_{p}$’s. The yellow line (consisting of four smaller straight lines at angles with each other) represents the effect of operating always (i.e., at any force ratio Χ) only a single one of the four gears, i.e., the one with the highest flux of ATP synthesis. 

### 2.5. Reproducibility and Accessibility of the Data

The Python script, the data used in these simulations, the visualization of the data, and all the figures are available on a publicly accessible Github https://github.com/YanfeiZhang1208/Gear-shifting-in-biological-energy-transduction (accessed on 27 June 2023).

## 3. Results

### 3.1. Mosaic Non-Equilibrium Thermodynamics and How the Variation of the Force Ratio Induces Gear Shifting of Catabolism

From the thermodynamic point of view, growth is driven by the Gibbs energy of catabolism ($\Delta G_{c}>0$); by our convention defined as chemical potential difference of substrates minus products). In catabolism, this Gibbs energy is partly converted to Gibbs energy in ATP (relative to ADP and phosphate; called $\Delta G_{p}>0$), whilst the rest is dissipated. This dissipation keeps the processes running at a high rate [27]. In anabolism, the ATP Gibbs energy is in part converted to Gibbs energy in biomass ($\Delta G_{a}$; positive for uphill growth). We first consider a single catabolic reaction fully coupled to the synthesis of *n*_1_ > 0 molecules of ATP. The remaining driving force $\left(\Delta G_{c}-n_{1}\cdot \Delta G_{p}\right)$ equals the catabolic Gibbs energy minus the stoichiometry (*n*_1_) multiplied by the Gibbs energy of ATP synthesis. As is customary in linear non-equilibrium thermodynamics [8,28,29], the flux is then assumed to be proportional to that remaining driving force. We use *L*_1_ for the proportionality constant (‘catalytic capacity’) and write for the dependence of its flux ($J_{c1}>0$) on the two relevant Gibbs energies:(1)$$J_{c1}=L_{1}\cdot \Delta G_{c}\cdot \left(1-n_{1}\cdot X\right)=\left(1-\varphi \right)\cdot L\cdot \Delta G_{c}\cdot \left(1-n_{1}\cdot X\right)\;$$
with the ‘force ratio’ X >0 representing the ratio of the Gibbs energy of ATP synthesis to the Gibbs energy of catabolism:(2)$$X\overset{\mathrm{def}}{=}\frac{\Delta G_{p}}{\Delta G_{c}}>0$$

We use linear (and even proportional) relations in this description, which is only an approximation of reality [30,31]. Our analysis will therefore only be able to illustrate tendencies in the behavior of the system, but this will suffice for the thrust of this article. The number of ATP molecules synthesized per unit (e.g., C-mole) catabolic process is denoted by stoichiometry *n*_1_, and the catalytic capacity of catabolism ($L_{1}>0)$ is written as fraction (1−φ) of a total catalytic capacity $L\overset{\mathrm{def}}{=}L_{1}+L_{2}>0$ of catabolism. The explicit mention of the stoichiometry $n_{1}$ reflects that we here use the more mechanistic ‘mosaic non-equilibrium thermodynamics’ of Westerhoff and Van Dam [27].

The expression shows that at high counteracting Gibbs energy of ATP synthesis ($\Delta G_{p}$), the rate of catabolism should be expected to decrease, a phenomenon related to ‘respiratory control’ [32] and ‘thermodynamic back pressure’ [27]. The flux of ATP synthesis ($-J_{p1}>0$), which is driven by $\Delta G_{c}$, is larger by a factor $n_{1}$:(3)$$-J_{p1}=\left(1-\varphi \right)\cdot n_{1}\cdot L\cdot \Delta G_{c}\cdot \left(1-n_{1}\cdot X\right)>0$$

In order to allow shifts between different pathways making ATP, we introduce a parallel pathway, again perfectly coupled to ATP synthesis:(4)$$J_{c2}=L_{2}\cdot \Delta G_{c}\cdot \left(1-n_{2}\cdot X\right)=\varphi \cdot L\cdot \Delta G_{c}\cdot \left(1-n_{2}\cdot X\right)$$
(5)$$-J_{p2}=n_{2}\cdot L_{2}\cdot \Delta G_{c}\cdot \left(1-n_{2}\cdot X\right)=\varphi \cdot n_{2}\cdot L\cdot \Delta G_{c}\cdot \left(1-n_{2}\cdot X\right)$$
with $L_{2}>0$ for its capacity and $n_{2}>0$ for its ATP synthesis stoichiometry. The fraction 1>φ>0 serves to indicate that this second pathway takes the remaining part of the total catabolic catalytic capacity:(6)$$\varphi \overset{\mathrm{def}}{=}\frac{L_{2}}{L}=\frac{L_{2}}{L_{1}+L_{2}}$$

The activity of the $n_{2}$ pathway is herewith φ/(1−φ) times higher than the capacity available to the former pathway. With the weighted-average stoichiometry defined by:(7)$$\nu \overset{\mathrm{def}}{=}\frac{n_{1}\cdot L_{1}+n_{2}\cdot L_{2}}{L_{1}+L_{2}}=n_{1}\cdot \left(1-\varphi \right)+n_{2}\cdot \varphi$$
the total catabolic and anabolic fluxes become:(8)$$\frac{J_{c}}{L\cdot \Delta G_{c}}=1-\nu \cdot X=\left(1-\varphi \right)\cdot \left(1-n_{1}\cdot X\right)+\varphi \cdot \left(1-n_{2}\cdot X\right)$$
(9)$$\begin{array}{cc} \frac{J_{c}}{L\cdot \Delta G_{c}} & =n\left(X\right)\cdot (1-\nu \cdot X)\\ & =\left(1-\varphi \right)\cdot n_{1}\cdot \left(1-n_{1}\cdot X\right)+\varphi \cdot n_{2}\cdot (1-n_{2}\cdot X)\\ & =n_{1}\cdot \left(1-n_{1}\cdot X\right)+\varphi \cdot (n_{2}\cdot (1-n_{2}\cdot X)-n_{1}\cdot (1-n_{1}\cdot X))\\ & =\left(1-\varphi \right)\cdot n_{1}\cdot \left(1-n_{1}\cdot X\right)+\varphi \cdot n_{2}-\varphi \cdot {\left(n_{2}\right)}^{2}\cdot X\\ & =\nu -((1-\varphi )\cdot {\left(n_{1}\right)}^{2}+\varphi \cdot {\left(n_{2}\right)}^{2})\cdot X \end{array}$$
Here, we defined the, now variable, ‘flux-ratio stoichiometry’ by:(10)$$n\left(X\right)\overset{\mathrm{def}}{=}\;\frac{-J_{p}}{J_{c}}$$

We also consider the possibility that there is a mechanism of uncoupling of the form of an ATP hydrolysis (‘leak’) reaction at flux $J_{p}^{\ell }$ that is independent of catabolism:(11)$$\frac{-J_{p}^{\ell }}{L\cdot \Delta G_{c}}=-\frac{L_{p}^{\ell }\cdot X}{L}=-\lambda_{p}^{\ell }\cdot X$$

Here, we again use the linear flow–force relation of linear non-equilibrium thermodynamics [8]. Both $L_{p}^{\ell }$ and $\lambda_{p}^{\ell }$ are positive. The flux–ratio stoichiometry n is a decreasing function of the force ratio X:(12)$$n\left(X\right)=\frac{n_{1}\cdot L_{1}+n_{2}\cdot L_{2}-\left({\left(n_{1}\right)}^{2}\cdot L_{1}+{\left(n_{2}\right)}^{2}\cdot L_{2}\right)\cdot X-L_{p}^{\ell }\cdot X}{L-\left(n_{1}\cdot L_{1}+n_{2}\cdot L_{2}\right)\cdot X}\;=\frac{\left(1-\varphi \right)\cdot n_{1}\cdot \left(1-n_{1}\cdot X\right)+\varphi \cdot n_{2}\cdot \left(1-n_{2}\cdot X\right)-\lambda_{p}^{\ell }\cdot X}{\left(1-\varphi \right)\cdot \left(1-n_{1}\cdot X\right)+\varphi \cdot \left(1-n_{2}\cdot X\right)}$$

For an example where the two stoichiometries $n_{1}\;\mathrm{and}\;n_{2}$ equal 0 and 1, respectively, the equation shows that the stoichiometry n(X) should decrease with increasing X.
(13)$${\left(n\left(X\right)\right)}_{n_{1}=0\;and\;n_{2}=1}=\frac{L_{2}\cdot \left(1-X\right)-L_{p}^{\ell }\cdot X}{L_{1}+L_{2}\cdot \left(1-X\right)}=\frac{1-\frac{L_{p}^{\ell }\cdot X}{L_{2}\cdot \left(1-X\right)}}{1+\frac{1-\varphi }{\varphi \cdot \left(1-X\right)}}$$

This is also evident if $L_{p}^{\ell }=0$, i.e., if the ATP synthesis is fully coupled to catabolism. It is this uncoupling-independent reduction in effective stoichiometry n(X) for large  X that we identify as gear shifting. For $L_{p}^{\ell }>0$ and $L_{1}=0;\varphi =1$, the equation shows the classical cause of the reduction of the effective stoichiometry, i.e., that due to uncoupling mechanisms independent of gear shifting catabolism:(14)$${\left(n\left(X\right)\right)}_{L_{1}=0\;and\;n_{2}=1}=1-\frac{L_{p}^{\ell }\cdot X}{L_{2}\cdot \left(1-X\right)}$$

Below, we shall see that the gear shifting itself can also introduce uncoupling.

The pink line in Figure 1a shows the hereby predicted dependence of the flux–ratio stoichiometry n(X) on the force ratio X for an example in which $n_{1}$ does not equal zero (but is still smaller than $n_{2}$) and there is no specific uncoupling mechanism ($L_{p}^{\ell }=0$). This decrease of the stoichiometry n(X) with increasing force ratio X derives from the phenomenon that with increasing force ratio, a smaller and smaller fraction of the phosphorylation flux flows through the high stoichiometry pathway (see Figure 1c). The total ATP synthesis flux decreases with increasing force ratio (black line in Figure 1a) and consequently, the flux–ratio stoichiometry n increases with the increase in ATP synthesis (and degradation when considering steady states) flux that occurs when the force ratio decreases (Figure 1b). Apparently, the behavior of the non-equilibrium thermodynamic system is driven by back pressure by the force ratio X.

Figure 1a also shows that the thermodynamic efficiency of the process, defined by [28,29]:(15)$$\eta \overset{\mathrm{def}}{=}\frac{-J_{p}\cdot \Delta G_{p}}{J_{c}\cdot \Delta G_{c}}$$
increases with increasing force ratio until it reaches a maximum, after which it decreases again. In the example, both the stoichiometry *n* and the efficiency η cross the abscissa at a force ratio of around 0.75. At this force ratio, the system is ‘stalling’, wasting Gibbs energy of catabolism whilst making no progress towards ATP synthesis. Beyond this force ratio, the efficiency and flux–ratio stoichiometry become negative, reflecting that ATP synthesis has turned into ATP hydrolysis. Figure 1c shows the fractional flux through pathway 1 and pathway 2 with two different ATP stoichiometries when varying the force ratio X. At low force ratio X, the fractional flux of the pathway with lower ATP stoichiometry increases with the increase of the force ratio X, which implies shifting from high gear to lower gear when increasing the force ratio X.

### 3.2. Phenomenological Non-Equilibrium Thermodynamics: Gear Shifting Affects the Phenomenological Stoichiometry as Well as the Degree of Coupling

ATP synthesis and catabolism are coupled to each other. This coupling has two aspects. First, with more intensive coupling, the ratio of the ATP synthesis flux ($-J_{p}$) to the catabolic flux ($J_{c}$) should increase. Second, with more intensive coupling, the catabolic flux should feel a stronger ‘backpressure’ by the Gibbs energy of ATP synthesis; for the case of mitochondria, this corresponds to the so-called ‘respiratory control’ [32]. This is expressed by the effect of the so-called ‘degree of coupling’ q on the following two equations [28]:(16)$$\frac{J_{c}}{L\cdot \Delta G_{c}}=1-q\cdot Z\cdot X$$
(17)$$\frac{-J_{p}}{L\cdot \Delta G_{c}}=q\cdot Z\cdot \left(1-\frac{Z}{q}\cdot X\right)$$
$\;J_{c}/L$ and $-J_{p}/L$ are the fluxes of catabolism and ATP synthesis, respectively, per unit catabolic catalytic activity (L). Z ≥ 0, 1≥q≥0, L>0, and X>0 represent the so-called phenomenological stoichiometry, the degree of coupling, the catalytic capacity of catabolism, and the ‘force ratio’ of the counteracting Gibbs energy of ATP synthesis to the driving Gibbs energy of catabolism, respectively. The two expressions are linear in the two Gibbs energies, i.e., L, Z, and q are assumed to be independent of $\Delta G_{c}$ and X. L will only change when the total activity of the system changes. Z and q will only change when the biochemical details of the coupling system change, such as the relative catalytic activities of the ATP synthesis enzyme and the catabolic enzymes (which will affect φ).

These two expressions do not describe reality precisely (reality is more nonlinear than this [27]), but they may still serve to illustrate qualitatively the roles played by the (Gibbs) energies, the phenomenon of ‘coupling’, the stoichiometry, the catalytic activity, and the thermodynamic ‘forces’. The equations show that if per unit catabolic process more ATP is made (as caused by an increase in the phenomenological stoichiometry Z), both the backpressure effect and the flux ratio should increase. However, stronger coupling and an increased such stoichiometry Z have opposite effects on the excess thermodynamic force (1−(Z/q)·X) through which catabolism drives ATP synthesis; the former increases and the latter decreases this force.

When mapping these phenomenological Equations (16) and (17) onto the above more mechanistic Equations (8) and (9) describing our system of two parallel pathways making ATP, we find for the phenomenological stoichiometry Z a special weighted average of the stoichiometries of the two pathways:(18)$$Z=\sqrt{\left(1-\varphi \right)\cdot {\left(n_{1}\right)}^{2}+\varphi \cdot {\left(n_{2}\right)}^{2}+L_{p}^{\ell }}$$

Notably, unless φ=0 or 1 this phenomenological stoichiometry differs both from the weighted average stoichiometry ν and from the flux–ratio stoichiometry n(X):(19)$$n\left(X\right)\overset{\mathrm{def}}{=}\;\frac{-J_{p}}{J_{c}}=\nu \cdot \frac{1-\frac{Z}{q}\cdot X}{1-\nu \cdot X}=Z\cdot \frac{q-Z\cdot X}{1-q\cdot Z\cdot X}$$

In addition, the degree of coupling *q* depends on the stoichiometries of the two pathways as well as on their relative activities, but in a different way:(20)$$q=\frac{\nu }{Z}=\frac{\left(1-\varphi \right)\cdot n_{1}+\varphi \cdot n_{2}}{\sqrt{\left(1-\varphi \right)\cdot {\left(n_{1}\right)}^{2}+\varphi \cdot {\left(n_{2}\right)}^{2}+L_{p}^{\ell }}}$$

Even at constant stoichiometries and with specific uncoupling, q and Z depend on each other through φ. If there is complete coupling, i.e., when there is only a single pathway, i.e., at q=1, i.e., complete coupling ($L_{p}^{\ell }=0$, and φ=0, or φ=1), *Z* equals the flux stoichiometry n(X), i.e., ratio of growth rate to catabolic flux. Even if there is no specific uncoupling, i.e., if $L_{p}^{\ell }=0$, coupling can be incomplete (q<1) because φ is neither equal to 1 nor to zero, i.e., because both pathways can operate. The flux stoichiometry is then in between $n_{1}$ and $n_{2}$, leading to a situation where both gears are operating at the same time. Then, one pathway synthesizes ATP whilst the other operates in reverse mode and hydrolyzes ATP. Coupling can also be incomplete when the stoichiometry n is precisely equal to $n_{1}$ or $n_{2}$, but then only if there is specific, gear shifting-unrelated, uncoupling such as due to a proton leak or growth rate independent maintenance ($L_{p}^{\ell }>0)$. 

More generally, the equation for the rate of ATP synthesis ($i.e.,-J_{p}=L\cdot \Delta G_{c}\cdot q\cdot Z\cdot \left(1-\left(Z/q\right)\cdot X\right)$) reflects that growth rate should be expected to increase with (*i*) the catalytic capacity of catabolism L, (*ii*) the magnitude of the Gibbs energy of catabolism (which is accordingly called the driving force for growth), (*iii*) the degree of coupling q (i.e., if there is less ATP diverted to maintenance or proton leakage and the coupling thereby increases, the growth rate should go up as well), and (*iv*) (at the lower force ratios, not at high force ratios where it will decrease due to increased effectiveness of the back pressure) the phenomenological stoichiometry Z. On the other hand, the growth rate should decrease (ν) with increasing back pressure from a higher Gibbs energy of ATP synthesis (or more generally, growth) relative to the Gibbs energy of catabolism.

### 3.3. Variations in the Phenomenological Stoichiometry Could Improve ATP Synthesis and Growth

The above mosaic non-equilibrium thermodynamic equations show that at any given catabolic Gibbs energy, the fluxes depend on five parameters, i.e., $X,\;\varphi ,n_{1},n_{2},\;\mathrm{and}\;L_{p}^{\ell }$. The phenomenological equations show that these parameters can affect the output characteristics of the system only through three parameters, e.g., Χ, q, and Z (or Χ, q, and ν or X, Z, and ν). Various authors have considered the variation of the output flux (here, ATP synthesis $-J_{p}$ or growth rate $-J_{a}$), yield ($Y=-J_{p}/J_{c}$), and thermodynamic efficiency ($\eta \overset{\mathrm{def}}{=}-J_{p}/J_{c}\cdot Χ$) with the Gibbs energy (‘force’) ratio [28]. They did this for fixed degrees of coupling equal to or smaller than 1 and fixed phenomenological stoichiometries Z. They also found that the ratio of output flux to input flux decreased from *Z* to zero (and even below) with increasing counteracting force ratio Χ. This is concordant with the purple curve shown for η in Figure 1a. Stucki also examined the dependence of the performance of mitochondrial oxidative phosphorylation on the degree of coupling q [29]. He considered the possibility of dual optimization, consisting of adjustments of both the force ratio Χ and the degree of coupling q so as to obtain optimality in terms of two criteria rather than one criterion. This seemed to explain incomplete coupling in mammalian mitochondria [29]. It could also underpin the notion that microbial growth may be optimized for both efficiency and growth rate [27,33].

Little attention has been paid to dependencies on the stoichiometry Z: being a ‘phenomenological’ stoichiometry, Z was considered immutable. In this paper, we break with this tradition and focus on ‘gear shifting’, i.e., on variations of the relative contributions to ATP synthesis of two pathways with varying stoichiometries causing variation in n(X), with n being the ratio between the ATP synthesis flux and the catabolic flux. Figure 2a shows how for three pathways that only differ in the magnitude of their phenomenological stoichiometry Z whilst having the same degree of coupling q, the ATP synthesis flux decreases with increasing counteracting force ratio Χ.

The green line in Figure 2a shows that at force ratios below 0.52, having the three pathways operate in parallel should be better than just operating one of them individually. In that range of force ratios, an ‘automatic’ tendency for the network to run flux through lower stoichiometry pathways (see Figure 1c), helps to keep the total flux of ATP synthesis high. At force ratios above 0.52, however, the network with all three pathways operative in parallel, turns to dismal performance due to the phenomenon that the pathway with the highest stoichiometry inverts its ATP synthesis flux into ATP hydrolysis. A futile, ATP hydrolyzing cycle hereby appears.

It should be better for the ATP synthesis if at the higher force ratios, the organism would switch between pathways, activating lower stoichiometry pathways whilst inactivating higher stoichiometry pathways. At fixed values of Χ and q, the ATP synthesis flux is a parabolic function of the phenomenological stoichiometry Z with a maximum (obtained by setting the derivative to zero) at:(21)$$Z_{maximal\;ATP\;synthesis\;flux}=\frac{q}{2\cdot X}$$
confirming that at small phenomenological stoichiometry (Z≪q/(2·X)), increasing this stoichiometry (Z) should enhance ATP synthesis, but not at high phenomenological stoichiometry (Z>q/(2·X)). Assuming that a regulatory network sets Z to this optimal value for each force ratio, the ATP synthesis flux should be given by:(22)$${\left(\frac{-J_{p}}{L\cdot G_{c}}\right)}_{optimal\;Z}=\frac{q^{2}}{4\cdot X}$$

The purple line in Figure 2a shows that indeed, such a ‘variomatic regulation’ of Z should produce a higher ATP synthesis flux than any other individual pathway with a fixed setting of Z. The corresponding purple line in Figure 2b shows how the variomatic flux–ratio stoichiometry decreases with increasing force ratio. Figure 2c shows that the flux–ratio stoichiometries increase with increase of ATP synthesis (reduction in force ratio). Notably, the situation leading to maximal ATP synthesis flux does not always lead to maximal yield. This is because also the catabolic flux varies with the force ratio.

### 3.4. Gear Shifting and Varied Relative Pathway Capacities Could Improve ATP Synthesis and Growth

In the above, we considered situations of different phenomenological stoichiometries at the same value of the coupling coefficient q. When gear shifting by changing the catalytic capacities of pathways with different stoichiometries, i.e., when gear shifting by changing the factor φ, most often the degree of coupling changes as well. Figure 3a plots the ATP synthesis flux versus the force ratio for various magnitudes of φ, at fixed magnitudes of the ATP stoichiometries $n_{1}$ and $n_{2}$ of the two pathways. Again, at low force ratios, the higher stoichiometry pathway produced more ATP synthesis than lower stoichiometry pathways, and at high force ratios (Χ > 0.25), this inverted. Now, however, as opposed to Figure 2a in which the degree of coupling was held the same for the different values of Z, all the lines intersect at a single point (Figure 3a), implying that, if the degree of coupling were allowed to vary, it would make no sense to gradually shift to lower stoichiometry with increasing force ratio. At one critical force ratio, it should be best to switch immediately from the highest to the lowest stoichiometry pathway (Figure 3a). A corollary would be that there be no evolutionary advantage to having multiple pathways with multiple stoichiometries. Then, two different stoichiometries should suffice to enable optimal adaptation of the ATP synthesis flux to the force ratio. Figure 3b,c shows the fractional flux through pathway 1 and pathway 2 with two different ATP stoichiometries when varying the force ratio X under different φ. Just as shown in Figure 1c, at low force ratio X, the fractional flux of the pathway with lower ATP stoichiometry increases with the increase of the force ratio X, which implies gears shifting from high to lower gear when increasing the force ratio X. With decrease of φ, the flux through pathway 1 increase.

Above, we showed that if two pathways take over the role of a single pathway, this has the effect of introducing a variable stoichiometry n(X). From this, we infer that having multiple pathways should enable an organism effectively to regulate its pathway stoichiometry. Figure 4 therefore shows the variation of the flux of ATP synthesis with the force ratio for four values of the stoichiometry n. Again, the ATP synthesis flux is best for high stoichiometry at low Χ and for low stoichiometry at high Χ. Like when we varied the phenomenological stoichiometry at constant degree of coupling, the lines do not all intersect at a single point. The implication is that it should make sense for the cells to shift their stoichiometries from 4 to 3, to 2 and then 1 as indicated by the line labeled ‘gear’ in the figure. The organism should do this by activating one pathway and suppressing all the others, again to prevent futile cycling. For this strategy, it is better to have a number of alternative pathways with different stoichiometries. This strategy corresponds to ‘automatic discontinuous gear shifting’ between integer gear settings, which differs from the ‘variomatic’ strategy, which is continuous.

## 4. Discussion

The study of gear shifting in cell growth is of interest because it has implications for understanding how cells optimize their metabolic pathways for efficient growth and energy transduction. The use of both mosaic and phenomenological non-equilibrium thermodynamics provides a comprehensive approach to investigating the underlying thermodynamic principles governing this process. The investigation of how the variation of the force ratio induces gear shifting from a mosaic non-equilibrium thermodynamics perspective is particularly relevant, as this approach allows for a more detailed understanding of the dynamics of individual components and their interactions within the metabolic pathways. In this paper, we showed that ATP synthesis flux decreases with increasing force ratio and that the gear setting (flux–ratio stoichiometry n) decreases with the decreasing ATP synthesis flux. In contrast, the thermodynamic efficiency of the process increases with increasing force ratio until it reaches a maximum and then decreases again.

By exploring the effects of gear shifting on the degree of coupling, we showed that both growth and ATP synthesis should always be expected to increase with the degree of coupling. The relation between growth (ATP synthesis) and phenomenological stoichiometry is more complex. At the lower force ratios, the growth/ATP synthesis will increase with phenomenological stoichiometry; at the higher force ratios, the growth will decrease with it.

The relationship between the phenomenological stoichiometry, gear shifting, and varied relative pathway capacities shows how they impact overall cellular metabolism. By studying the effects of these factors on ATP synthesis and growth, the study shows that ATP synthesis flux is high for high stoichiometry at low force ratio and for low stoichiometry at high force ratio.

In this study, we used two types of non-equilibrium thermodynamics. One enabled us to discuss the performance of cell growth as free-energy transducer with the least possible detail on mechanism. This enabled us to show the effects of variation of the degree of coupling and the phenomenological stoichiometry. As its name implies, Z is a phenomenological stoichiometry, however, and it is unclear how gear shifting between catabolic pathways with different ATP synthesis stoichiometries, should affect Z. For this, we resorted to mosaic non-equilibrium thermodynamics [10], which enabled us to insert two parallel catabolic pathways with different ATP stoichiometries. This showed that *Z* should become a weighted function of the activities and stoichiometries of the two pathways but would remain independent of the counteracting force ratio. Only the actual stoichiometry *n* between ATP synthesis by catabolism as a whole and the total catabolic flux became dependent on the counteracting force ratio. The more thermodynamically uphill growth would be, the lower *n*, corresponding to a shift to lower gear setting. Further analysis suggested that by regulating which of various parallel catabolic pathways they use, organisms may improve their growth rate both at low and at high counteracting thermodynamic force.

In using the two types of non-equilibrium thermodynamics, rather than a complete and validated kinetic or flux balance constrained model [34], the present study appears to be subjected to a strong limitation. The two types of thermodynamics both assume linear and even proportional relationships between fluxes and Gibbs energy differences, and although there is experimental evidence for appreciable linearity in some cases [31,35], that linearity is not often a proportionality, nor is it found generally. The reason why we still took to these non-equilibrium thermodynamic descriptions is twofold. First, a complete and validated nonlinear kinetic description of microbial growth is not yet available. This is understandable, as such growth is a function of the activities of hundreds of enzymes [36]. Second, few kinetic models are explicit on the thermodynamic implications they have and are subject to [37,38]. Third, although flux balance analysis models do accommodate the participation of hundreds of enzymes, they lack thermodynamics [39,40]. Consequently, we had to restrict our analysis to the existing linear non-equilibrium approaches.

It is our expectation, however, that this restriction does not invalidate the major conclusions of the present paper, i.e., that organisms may shift between catabolic pathways that have different ATP stoichiometries. First, linear non-equilibrium thermodynamics has been able to develop insights on a variety on important phenomena in biology, including self-organization [7], stability [7,27,41], and oscillations [42,43]. Second, there is ample experimental evidence that organisms do vary the relative expression level of parallel catabolic routes with different ATP stoichiometries [23,24,44]. Third, after all, there is ample evidence that various cell types vary their utilization of catabolic routes that yield highly different amounts of ATP; this includes the Warburg and WarburgQ effects [45].

The present paper may help improve on the omission of thermodynamics from many FBA and kinetic models in the future, as it shows how the ability to shift between parallel pathways with different ATP stoichiometries that can be identified by FBA (and flux variability analysis) (Mondeel et al. [21]) is an aspect that evolutionary optimization may be driven by. It may well provide another partial explanation of why most microorganisms grow at yields and thermodynamic efficiencies far below what should be possible theoretically [33,46], or have metabolic capacities well above those minimally needed [47].

Overall, the study of gear shifting presented in this paper may improve our understanding of the fundamental thermodynamic principles that govern cell metabolism.

## Figures and Tables

**Figure 1 entropy-25-00993-f001:**
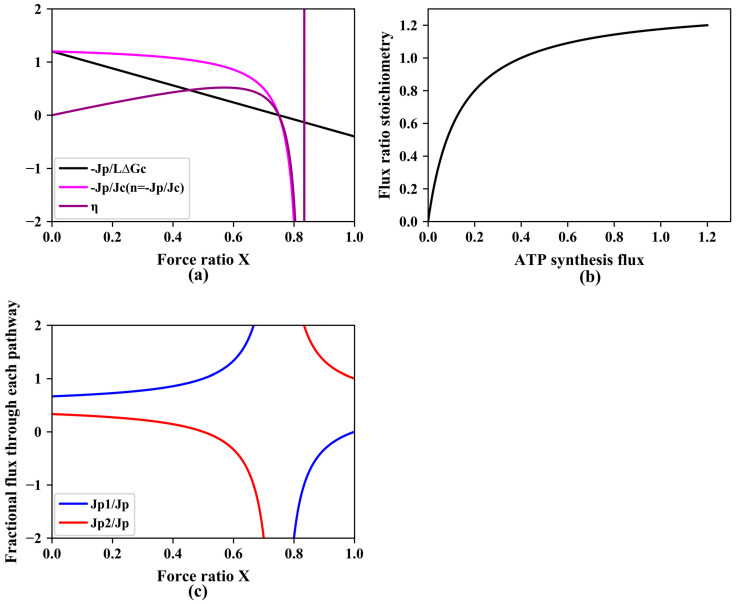
(**a**) Dependence of ATP synthesis flux ($-J_{p}$; normalized by maximal catabolic flux,$\;L\cdot \Delta G_{c}$), variable flux–ratio stoichiometry n = $(-J_{p}/J_{c}$) and thermodynamic efficiency (η) on the counteracting force ratio $X=\Delta G_{p}/\Delta G_{c}$. Equation (9) was used for the simulation of $-J_{p}/\left(L\cdot \Delta G_{c}\right)$ as function of force ratio X; Equations (8) and (9) were used for the calculation of $-J_{p}/J_{c}$; the equation $\eta =-(J_{p}/J_{c})\cdot X$ was used for simulation of η as function of force ratio X. (**b**) Flux–ratio stoichiometry versus ATP synthesis. Increase of the variable stoichiometry n= ($-J_{p}/J_{c}$) with increasing ATP synthesis flux. At a force ratio of 0.75, the output flux of ATP synthesis reverted to ATP degradation, while catabolism continued. This corresponds to a car still using gasoline to try to move forward and upward but being forced back down due to gravity. Consequently, the stoichiometry and the thermodynamic efficiency become negative. At the force ratio of 0.83, catabolism also inverted. At this and higher force ratios, the model simulates reverse operation where ATP hydrolysis would drive reversal of catabolism, which is not often realistic. (**c**) Fractional flux through two pathways when varying the force ratio X. Equations (3) and (9) were used for the calculation *of*
$J_{p1}/J_{p}$; Equations (5) and (9) were used for the calculation of $J_{p2}/J_{p}$. Results of computations for two parallel catabolic pathways with φ = 0.2, $\;n_{1}$ = 1, $n_{2}$ = 2, L = 1, $\Delta G_{c}$ = 1, $L_{p}^{\ell }=0$.

**Figure 2 entropy-25-00993-f002:**
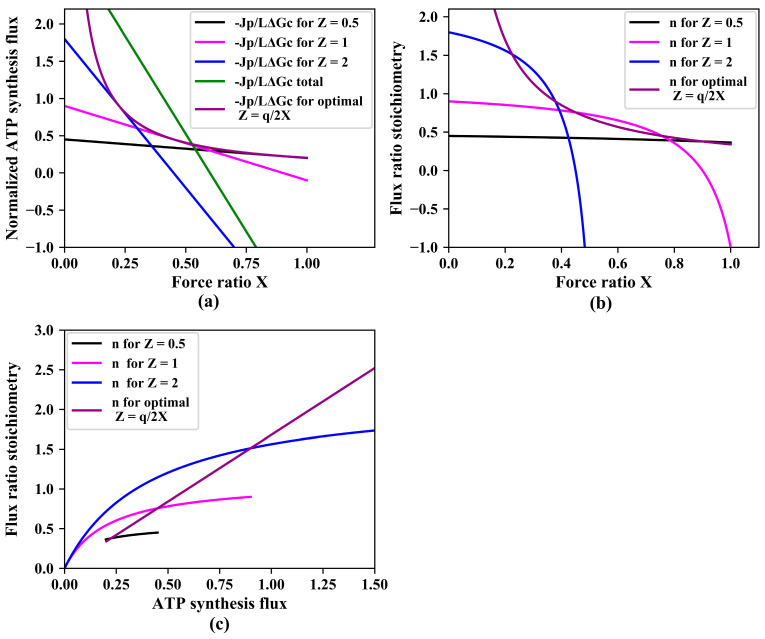
Gear shifting. (**a**) Normalized ATP synthesis flux versus force ratio at three magnitudes of the phenomenological stoichiometry Z, as well as (green line) their sum total and (purple line) ATP synthesis for ‘variomatic gear shifting’ optimal with respect to maximal ATP synthesis flux. Equation (17) was used for calculation of $-J_{p}/\left(L\cdot \Delta G_{c}\right)$ as a function of force ratio X. In these simulations, q = 0.9 and various values of Z (0.5, 1, 2, and the optimal variomatic $Z=q^{2}/X$) were used as indicated. $-J_{p}/\left(L\cdot \Delta G_{c}\right)$*_total_* was calculated as the sum of three. (**b**) Flux ratio stoichiometry as a function of the force ratio. Equation (19) was used for simulation of n. (**c**) Flux ratio stoichiometry versus ATP synthesis flux.

**Figure 3 entropy-25-00993-f003:**
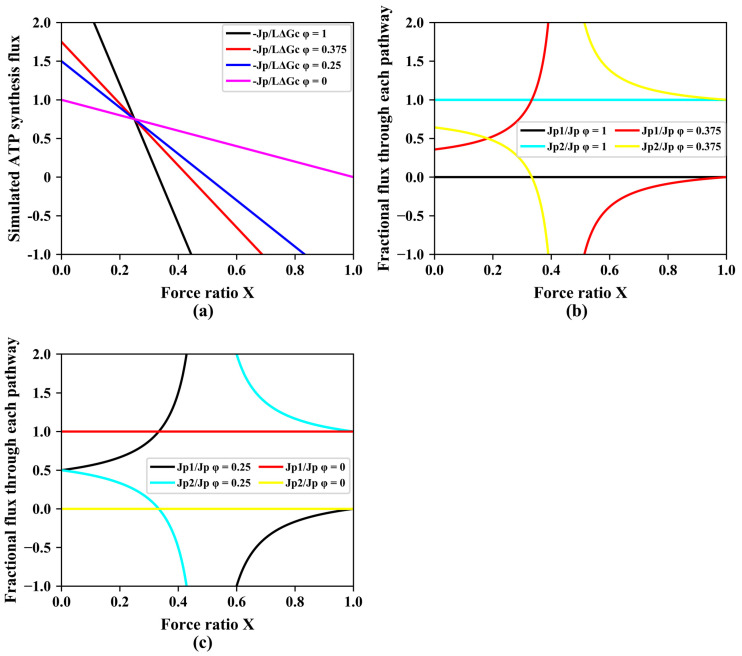
Gear shifting of different pathways at different relative pathway activities φ. (**a**) Simulated ATP synthesis flux through a dual pathway (pathways 1 and 2 with two different ATP stoichiometries) as function of force ratio at various relative pathway capacities. (**b**) The fractional flux through pathways with two different ATP stoichiometries when varying the force ratio X under different φ’s (1, 0.375). (**c**) The fractional flux through pathways with two different ATP stoichiometries when varying the force ratio X at different φ (0.25, 0). In these simulations, both Z and q varied as a function of φ, which was kept constant at any of four values, while Χ was varied. Equation (9) was used for the simulation of $-J_{p}/\left(L\cdot \Delta G_{c}\right)$ as function of force ratio Χ. Equations (3) and (9) were used for the calculation of $J_{p1}/J_{p}$; Equations (5) and (9) were used for the calculation of $J_{p2}/J_{p}$. In these simulations, $n_{1}$ = 1 and $n_{2}$ = 3, L = 1, $\Delta G_{c}$ = 1, $L_{p}^{\ell }=0$, whilst for (**a**) four values of φ (1, 0.375, 0.25, and 0) were used. φ is the catalytic activity of pathway 2 as compared to the catalytic capacity of the two combined.

**Figure 4 entropy-25-00993-f004:**
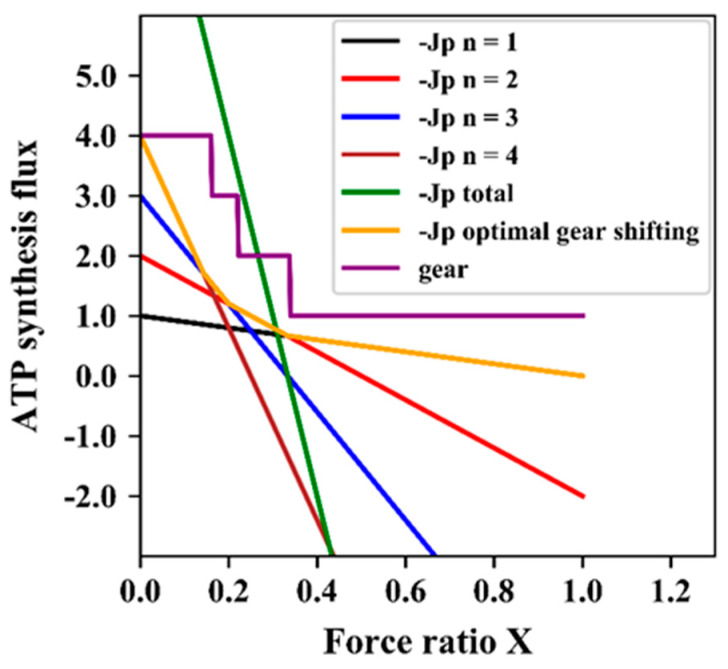
Discontinuous optimal gear shifting. ATP synthesis as function of counteracting force ratio for four different stoichiometries, all together, as well as the optimal gear shifting case with the corresponding gear settings. The equation $-J_{p}=\left(1-\varphi \right)\cdot n\cdot L\cdot G_{c}\cdot \left(1-n\cdot X\right)$ was used for simulation of $-J_{p}$ as function of force ratio Χ at four fixed values of n. In these simulations, $n_{1}$, $n_{2},n_{3},$ and $n_{4}$ were taken equal to 1, 2, 3, and 4, φ = 0, L = 1, $\Delta G_{c}$ = 1. $J_{p\;total}$ is the sum of the four $J_{p}$’s. The yellow line (consisting of four smaller straight lines at angles with each other) represents the effect of operating always (i.e., at any force ratio Χ) only a single one of the four gears, i.e., the one with the highest flux of ATP synthesis. The difference of the optimal gear setting with that of Figure 2a is that the shifting is not continuous but only between integer values of Z as shown by the purple line labelled ‘gear’.

## Data Availability

No new data were created or analyzed in this study. Data sharing is not applicable to this article.

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
