# Peer review of "Gear Shifting in Biological Energy Transduction"

_entropy, 2023, doi:10.3390/e25070993_

Round 1

Reviewer 1 Report

In the paper authors have developed the theory behind gear shifting in cell growth processes in an orginal and creative way. For this purpose, they used mosaic and phenpmenological nonequilibrium thermodynamics. They explored how shifting gears can be beneficial to an organism when growth becomes thermodynamically difficult. The work is elegant and has important cognitive and applied values. I recommend this paper for publication in the journal Entropy.

Author Response

Dear reviewer,

Thank you very much for the recommendation of our paper for publication in this journal. 

Sincerely yours

Yanfei Zhang and Hans V. Westerhoff

Reviewer 2 Report

The authors try to underpin biological gear shifting and identify possible mechanisms by utilizing existing non-equilibrium thermodynamic theories. This work theoretically shows that gear shifting differs from altering the degree of coupling and that living systems may use it to optimize their performance. The results are interesting and inspiring. However, in my opinion, this work would only be suitable for publication in Entropy following significant revisions.

Major Concerns:

(1)The second section, "Materials and Methods", contains numerous equations with insufficient explanation of the variables and derivations. These explanations are deferred until the third section, which may confuse readers. Moreover, the rationale for adopting certain equations (for instance, eqs.(1) and (18)) is not clear. Detailed explanations are critical to elucidate the theory proposed, especially for readers not familiar with "mosaic non-equilibrium thermodynamics". 

(2) The self-citation rate of this work is as high as 40%, which may be not acceptable. 

(3) The author provide several pathways to demonstrate the "gear shifting" phenomenon, however only weight-averaged stoichiometry or weight-averaged flux is shown. It would be more clear if the percentage of each pathway is plotted to show that the "gear" is shifting.

Minor Concerns:

There are several typos in this manuscript. For example, in line 160 of eq.(9), half of the equation is missing. And in line 240, the number of reference is wrongly placed.

Round 2

Reviewer 2 Report

The authors have addressed my concerns. The manuscript is now in a publishable state in its present form.